# The implementation gap: Cross-sector management of heat-related health risks in Western cape, South Africa

Amanda V. Quintana[1]*, Lucy Gilson[1,2], Sari Kovats[1], Caradee Y. Wright[3], Susannah H. Mayhew[1]

**1** Faculty of Public Health Policy, London School of Hygiene and Tropical Medicine - Keppel Street, London, United Kingdom, **2** Department of Public Health and Family Medicine, University of Cape Town; London School of Hygiene and Tropical Medicine - Observatory, Cape Town, South Africa, **3** Environment and Health Research Unit, South Africa Medical Research Council - Francie van Zijl Drive Parowvallei, Cape Town, South Africa

* Amanda.Quintana@lshtm.ac.uk

## Abstract

As temperatures rise due to climate change, so do adverse health effects. In response, many countries, including South Africa, have developed heat health action plans to address these threats to public health. In the Western Cape province, increasing heat events necessitate a well-coordinated response across governance levels and sectors. Understanding how heat risks are governed, particularly at sub-national and local levels, is critical for safeguarding public health and building resilience to future climate challenges. This study draws on 31 in-depth interviews and cross-references a previous policy document analysis to examine how South African decision-makers, both within and outside the health sector, at the Western Cape provincial and municipal levels, manage heat-related health risks. Using an adapted Multiple Governance Framework, the analysis investigates how subnational and local stakeholders work to manage heat-related health risks, some of which are aligned with South Africa's 2020 Heat-Health Action Guidelines. The findings reveal that despite the existence of the Action Guidelines and recognition among Western Cape decision-makers of the urgency of heat-related health risks, implementation remains fragmented. While provincial and municipal stakeholders are actively working to mitigate the health impacts of extreme heat, subnational and local actors were not involved in developing the Heat-Health Action Guidelines limiting their applicability at the local level. The analysis further highlights governance challenges and opportunities that emerge across system, organizational, and individual scales, emphasizing the significant role of decision-makers' perceptions in shaping responses. Strengthening coordination, defining departmental roles, and enabling local adaptation of policy strategies will be essential for improving heat-health action. By addressing these governance gaps, decision-makers in the Western Cape can manage current and

**Data availability statement:** Yes - all data are fully available without restriction; All relevant data are within the manuscript and its Supporting Information files.

**Funding:** This research study did not receive research funding. A travel grant was offered in 2022 by LSHTM to support partial field work costs in South Africa (LSHTM Travel grant to AVQ). The funders had no role in study design, data collection and analysis, decision to publish, or preparation of the manuscript.

**Competing interests:** The authors have declared that no competing interests exist.

future heat-related health risks and communities can be better equipped to withstand the increasing frequency and intensity of extreme heat events.

## Introduction

As temperatures rise due to climate change, so do adverse health effects. Extreme heat events are associated with increased mortality and hospitalization, particularly among vulnerable populations [1–3]. Research in South Africa has identified a notable association between hot days and increased mortality, especially among children and older adults, who are particularly vulnerable to heat-related health impacts [4,5]. Realizing the significance of increased heat risk on human health and that most adverse health effects of heat are preventable, the World Health Organization issued guidance to develop heat-health action plans, listing actions that should be taken at different levels and sectors [6]. Managing heat risks for health therefore entails developing and implementing coordinated interventions, such as heat early warning systems, targeted public health campaigns, and tailored support for vulnerable populations, to reduce morbidity and mortality, especially during extreme heat events [7,8].

Research on the governance of heat risks for health is especially important as climate change intensifies the frequency, duration, and severity of heat waves. Although peer-reviewed literature on heat-related health governance is in the early stages, several studies indicate that implementation is limited and national heat health action plans require improved structure and coordination [9–12]. Clear governance structures and policies to effectively adapt and prepare for inevitable and inequitable climate hazards, such as increased heat, are essential to cross-sectoral coordination and long-term planning, accountability, and ultimately, resilience [13,14]. A study on urban heat governance in 22 large cities found public health institutions are actively engaged in extreme heat planning, yet their involvement is often limited to what Sheehan *et al* term 'reactive measures', like issuing heat warnings and risk communications [15]. In cases where public health agencies are more engaged, they take on more complex activities like preparedness and vulnerability mapping to address heat-related health impacts.

Beyond the engagement of the health sector in heat planning, the implementation of health adaptation actions to heat has been challenged by a lack of local authority engagement, particularly in heat action plan development [12]. This is crucial because as empirical studies suggest, like emergency management, the management of heatwave risks is increasingly decentralized, and implementation relies heavily on subnational and local stakeholders who understand and are best placed to respond to local needs [12,16]. Understanding interactions between stakeholders reveals how decisions made across sectors at the national, regional, and local levels interconnect to shape policy implementation and action [17,18]. It also provides a better understanding of the relationships between government levels across mitigation and adaptation policy issues. Climate governance scholarship has long acknowledged the criticality of decisions made at regional and local levels for an effective response, [19,20]. This cross-level coordination is particularly essential for managing

heat-health risks, that demand synchronized efforts among multiple sectors and scales to design and deliver adaptive heat strategies that address diverse population needs and mitigate adverse health impacts.

Studies examining national 'heat-health action plans' indicate that such plans assign roles and responsibilities in a "top-down" approach that do not permit collaboration with or motivate subnational stakeholders, especially healthcare staff, whose involvement is necessary to carry out heat-related health actions [10,12,21,22]. A systematic review of implemented adaptation actions to extreme heat shows that heat governance varies across countries, where high-income nations primarily treat heat as a public health issue, whereas low- and middle-income countries focus on agriculture and livelihood-based impacts, reflecting broader socioeconomic vulnerabilities [23]. However, in South Africa, a middle-income country with high inequity, a heat health action plan explicitly frames heat as a health issue, emphasizing the country's recognition of heat-related health risks and the need for cross-sectoral coordination to manage them effectively.

## Heat-health policy and governance in the Western Cape Province

In South Africa the 2020 National Heat-Health Action Guidelines (Action Guidelines, 2020), developed by the health sector, lists a series of actions that must take place to protect public health from extreme heat and increasing temperatures. The Action Guidelines mandate that provinces and municipalities develop heat health plans informed by the Guidelines, with the health sector designated as the lead agency responsible for these heat health plans across all levels of government [24]. However, a previous review of the Action Guidelines, along with other national and Western Cape provincial climate and health policies, found that the Guidelines lacked sufficient elements to support effective implementation of health adaptation measures [25]. Specifically, the study identified gaps in the Action Guidelines, including the absence of allocated resources, a monitoring and evaluation framework, and clear information on the allocation of responsibilities and timelines for carrying out heat-related health risk activities [25]. This study assesses these gaps further.

Before the South Africa Heat-Health Action Guidelines (2020) were developed, key climate adaptation documents already addressed health adaptation measures related to heat and heat planning. For example, both the National Climate Change Response Policy (2011) and the unpublished second-iteration of the South Africa Climate and Health Adaptation Plan (2020) included measures such as developing and implementing a Heat-Health action plan, as well as initiatives to raise public awareness about the health risks of heat, outlining response actions the public should take to protect their health.

The Western Cape province faces increasing climate-related challenges, including droughts, wildfires, and regular heatwaves, which heighten heat-related health risks and necessitate a well-coordinated response across governance levels and sectors. These challenges unfold within the context of South Africa's three-tier health system, where the national government sets policy and allocates funding to provinces, which are responsible for policy implementation and service management. In the Western Cape's metropolitan municipality, the City of Cape Town, health service delivery is shared between the City's health department and the provincial department of health, creating parallel responsibilities around primary health care. The Western Cape province also incorporates five district municipalities which, with the City of Cape Town, are responsible for providing environmental health services. Additionally, the province has 24 local municipalities that focus on essential services, such as water and sanitation and waste management. The combination of climate-related challenges and a multi-level health and governance system makes the Western Cape an interesting context for understanding the complexities of implementing heat-mitigation strategies and managing heat-related health risks.

Previous work looking at the coherence of climate and health policies across sectors in South Africa identified some linkages between national policies relating to measures to mitigate heat-related health risks [25]. Although the Western Cape's provincial policies did not include heat-health measures as national policies did, municipal policies, like the City of Cape Town's Climate Change Strategy and Action Plan, include specific actions. These actions include the development of heatwave and high-heat day action plans and standard operating procedures, the development and implementation of a network of cooling centers, and the development and implementation of an early-warning and real-time monitoring

system for heat [26]. The City of Cape Town is in the early stages of implementing the Heatwave and High-Heat Day Action Plan approved in late 2023 that aims to address immediate risks of extreme heat [27]. The City continues to work on a comprehensive governance structure to manage urban heat risk, where it is acknowledged that the involvement of the health sector, including engaging health workforce and improving public health messaging, is critical for operationalization of the Action plan [27]. The existence of such policies suggests that decision-makers in the Western Cape and City of Cape Town are aware of heat-health policies and are actively managing heat-related health risks.

Although South Africa has formally recognized heat as a health issue and developed national action guidelines, little is known about how heat-related health measures are translated into practice at subnational and local levels, the challenges encountered, and the actions taken to manage such risks. This paper investigates how South African decision-makers across governance levels and especially at the Western Cape provincial and local levels work to manage heat-related health risks and identifies the challenges and drivers of action. Building from a previous study that reviewed the country's Heat Health Action Guidelines, the findings from this study suggest there are gaps in implementation and action occurring on managing heat risks beyond the policy itself. Furthermore, the study advances our understanding of the governance of heat-related health risks, which is an increasingly urgent research area as global temperatures continue to rise.

## Methods

### Data collection and interview participants

This study is part of a larger research project that aimed to understand the factors that influence the implementation of climate adaptations for health in Western Cape, South Africa by assessing responses to drought and heat. Study participants were selected based on the following criteria: senior decision-makers from South Africa, either within or outside the health sector, primarily from the Western Cape, but also from municipal and national levels, who have the potential to influence climate change adaptation efforts for health. Participants were identified and invited to participate in 50–90-minute interviews through both policy document authorship and a snowball sampling technique, where initial contacts recommended additional relevant stakeholders.

Prior to interviews, an interview guide was developed, piloted, and validated by the Supervisory team. A total of thirty-one in-depth interviews were conducted, both in-person and online, between February and June 2022. Participants represented three levels of governance, national, provincial, and local, and comprised of health sector stakeholders as well as non-health sector participants from the environment, disaster response, and development fields (see **Table 1**).

Participants were invited to interview over email and received a project information sheet and consent form. Participants signed a consent forms prior to interview sessions, allowing recording, notes, and anonymized short quotes to be made. The use of quotes in this paper are of individual participants that include their governance level and whether they are in or outside of the health sector. Ethical approvals were received from the University of Cape Town (#756/2021) and London School of Hygiene and Tropical Medicine (#26533).

**Table 1. Characteristics of Participants.**

| Participant Type | Number of Participants |
|---|---|
| **By Governance Level (**n = 31**)** | |
| National | 8 |
| Provincial | 16 |
| Municipal | 7 |
| **By Sector (**n = 31**)** | |
| Health Sector | 15 |
| Non-Health Sector | 16 |

Interview sessions were audio-recorded and guided by a piloted interview guide where participants were asked how they manage climate risks, particularly heat risks, if they are familiar with a list of climate and health policies at national, western cape, and municipal level, such as the 2020 National Heath Health Action Guidelines, and if and how these policies are implemented. The full interview guide can be found in S1 File.

## Analysis

Interviews were transcribed verbatim, anonymized to protect participant identities, and imported into NVivo 10 for analysis. An inductive and deductive thematic analysis approach was employed using a code frame derived from the interview guide and a set of interviews. During the initial round of coding, various codes were then deductively applied, including 'Engagement with Policy' and a primary code for 'Heat.' In a subsequent round of coding, analysis of the 'Heat' code revealed two sub-codes: 'Actions/Responses' and 'Policy and Planning.' As interviews explored another climate risk, drought, only information related to discussions on heat were collected from interviews and included in the analysis.

To systematically capture participant perspectives, two spreadsheets were created, one for each sub-code, organizing data by participant governance level and sector. The data were then analyzed inductively to identify key themes within these two sub-codes. The Hill and Hupe's Multiple Governance Framework [18] governance action domains and action scales (discussed below) were also applied to gain deeper insights into implementation. The key themes are discussed within the context of the framework presented in the results section.

Participant information was also compared with findings from a previous policy document analysis, that outlined heat-health action measures in the National Heat Health Action Guidelines [25] to determine which measures participants were actively working on. This involved cross-referencing the health adaptation measures listed in the Action Guidelines with the activities participants reported they were engaged in or familiar with to address heat-related health risks. Finally, during interviews participants were asked to indicate their awareness of the Action Guidelines, and this information was noted for each participant during analysis.

## Adapted multiple governance framework

Hill and Hupe's [28] analytical 'Multiple governance framework' (MGF) offers specific insights into the practice of governance, with a focus on decision-making in policy implementation. Hill and Hupe [18] note the importance of looking at both the 'what' and the 'how' of governance which includes understanding the authority of national structures as well as how collective action and processes are enabled across governance levels. Hill and Hupe's MGF is a conceptual tool whose specific elements can be applied to derive deeper insights into how stakeholders across governance levels and sectors work to manage heat-related health risks.

Hill and Hupe's [18] concept of the "action level" is central to the MGF and refers to different types of governance decisions. These include **constitutive governance,** representative of the South African national level and which addresses decisions about rules within the broader politico-administrative system; **directional governance**, which encompasses decisions on policy directions, strategies, and planning at both a government-wide level and within specific sectors, representative of all levels of government; and **operational governance**, which represents the provincial and local level and involves decisions related to the implementation of policies and priorities. Hill and Hupe's "action scales" consists of three levels: the **system** scale, which focuses on macro-level mechanisms, sectoral and cross-sectoral planning, and responsibilities for the whole system; the **organizational** scale, which examines the interactions between departments, institutions, and the collaboration required to implement priorities; and the **individual** scale, which refers to decision-making and behaviours at the micro-level, focusing on individual stakeholders and their actions in specific settings.

An alternative widely used health policy process framework developed by [29] elaborates on the importance of how an issue is framed and understood. The ideas decision-makers have about heat risks can can shape political support and influence collective action [29]. Drawing from this framework, the individual scale in the MGF has been further adapted to

**PLOS Global Public Health**

include how individual perceptions influence stakeholders' understanding and actions on policy issues. Together, these scales and levels constitute the nine interlinked governance action domains within Hill and Hupe's MGF, elaborated in the study's findings and seen in **Table 2**.

## Results

The findings primarily reflect the perspectives of provincial and municipal participants (n = 23), with national level perspectives (n = 8) included to provide additional context, such as describing activities coordinated with local levels and offering insights on heat risks. The findings are presented in two sections. A first short section captures the extent to which provincial and municipal stakeholders were aware of the Heat Health Action Guidelines and took action on heat adaptation measures aligned then, participants' experiences in managing heat-related health risks are discussed. The findings are presented using the adapted MGF framework, considering system, organizational, and individual levels.

### Awareness of and action on heat-health measures

Of provincial and municipal participants, 78% (n = 18) revealed they were unaware of the National Heat-Health Action Guidelines when asked during interviews. Like this provincial non-health sector participant expressed: "*The National Heat Health Action Guideline I didn't even know existed.*" P005 Provincial, Non-Health.

This suggests that subnational and local decision-makers were not involved in the Action Guideline development process and likely lacked clarity on their roles and responsibilities in executing the plan. Among the few aware (n = 5), most reported limited action on the outlined strategies. Although not asked about specific actions, participants, particularly health sector participants, identified some measures to address heat-related health risks that aligned with the Action Guidelines, such as the retrofitting of health facility waiting rooms and communicating public warnings of increased heat, particularly for schools.

The National Heat Health Action Guidelines lists thirteen actions, or health adaptation measures, comprised of five short-term, five medium-term, and three long-term actions. Out of the thirteen actions, participants spoke of heat-health measures aligned with just three actions [24]:

1) "Review healthcare facilities' thermal comfort" (medium-term)

2) "Assess the heatwave definition and finalize a set of messages about heat and health" (short-term) and

**Table 2. Adapted Multiple Governance Framework.**

| Action Level →<br>Action Scale ↓ | CONSTITUTIVE GOVERNANCE<br>Rulemaking and Laws | DIRECTIONAL GOVERNANCE<br>Policy and Strategy development | OPERATIONAL GOVERNANCE<br>Implementation and action |
|---|---|---|---|
| **SYSTEM**<br>Macro level mechanisms, policy and planning | **Institutional design** | **General rule setting** | **Managing trajectories** |
| **ORGANIZATIONAL**<br>Meso level<br>Government Departments and Facility Infrastructure | **Designing (inter-) organizational settings** | **Context Maintenance** | **Managing Relations** |
| **INDIVIDUAL**<br>Micro level<br>Decision-makers/ Frontline workers<br><br>**PERCEPTION**<br>Understanding and Framing of Issue | **Internalization of values & norms** | **Situation-bound rule application** | **Managing contacts** |

**Source:** Adapted from Hill and Hupe (2009 updated 2016).

3) "Activate heat-health warnings alongside heatwave alerts" (short-term)

One participant expressed that the lack of awareness could be explained by the COVID-19 pandemic, which coincided with the plan's development, affecting the plan's dissemination.

"*Some of these documents or policies that have been developed during like COVID. They haven't really had much traction because of the limited engagement*." P002, Provincial, Non-Health Sector.

Although unaware of the plans, stakeholders were still addressing heat-related health risks. A health participant who worked closely with disaster management shared that although unfamiliar with the Action Guidelines at the time, actions to address heat were underway, specifically, to develop a heat index with other departments such as agriculture, education, disaster management, including the South African Weather Service.

"*So, [what] we want to [do] now, along with the colleagues with disaster management, with agriculture, education, [is] to develop a heat index*" P008, Provincial, Health Sector

**Experiences managing heat-related health risks**

To explore the challenges and opportunities that provincial and municipal stakeholders faced in managing heat-related health risks, this study applied the systems, organizational, and individual action scales. As highlighted in the preceding section, awareness and engagement with the Action Guidelines were minimal among provincial and municipal participants. However, despite this limited awareness, some planning actions were undertaken to address heat-related health risks at various governance levels.

**System level (Marco-level Mechanisms and Dynamics).** The themes that emerged at the systems level relate to collaboration and coordination across multiple sectors and correspond with the framework domains *general rule setting* that is about setting policy directions and *managing trajectories* that can explain how priorities are operationalized.

Participants emphasized that heat is a complex and cross-cutting risk that does not fall neatly under one sector's remit and requires collaboration across sectors and governance levels to ensure an effective response. When speaking of work to address the health impacts of heat one participant highlighted how collaboration between the health and environment sectors during a past heatwave was possible because heat was framed as a shared challenge, encouraging sectors to work together.

"*For the first time we had the heat wave seen as one enemy that all of us are against and work against. And that is what happened in COVID, the same. And I think that is the right approach.*" P004, Provincial, Health Sector

The shared challenge of addressing a heatwave brought together stakeholders from multiple sectors, exemplifying the principles of *general rule setting*. At the systems level, the *managing trajectories* domain highlights how multisectoral innovation and cross-sector coordination are essential for effectively managing heat-related health risks. Cross-sector coordination can drive innovative approaches to heat response. A municipal participant touched on this when describing how the management of heat risks differs from managing drought, requiring adaptive strategies and strong collaboration to ensure an effective response.

"*The city government has a huge control over the way it responds to drought, it has very minimal control over the way it responds to heat, at least in the short term. So, heat response requires a lot more ongoing engagement with ordinary citizens about how people can protect themselves, and how they can protect vulnerable people around them... so heat has a very different response. And we need to do a lot more work in this regard.*" P024, Municipal, Non-Health Sector

Finally, at the time of the interviews, both health and non-health sector participants across governance levels discussed plans to develop a heatwave index that would establish thresholds, protocols, and public messaging. Along with a provincial cross-sectoral heat index that was inspired by an existing fire index that facilitates coordination through a dedicated forum (P008) a municipal-level initiative was proposed that addresses gaps in current forecasting tools which often complicate action planning, particularly for the most vulnerable populations, because they are not locally specific.

"*So, we want to develop a heat monitoring network of devices across the city so that we know what's happening in heat and high-risk areas, because that's not what the weather forecast tells you, it tells you what's going to happen regionally.*" P012, Municipal, Non-Health Sector

**Organizational level (government departments and health facilities).** Addressing heat risks also requires targeted actions by government departments and health facilities that play a crucial role in implementation. The organizational level considers opportunities and challenges faced by these organizations in addressing heat-related health risks that include adapting strategies to local needs and associated costs, which aligns with the *context maintenance* domain emphasizing that policies should be tailored to specific contexts. Similarly, the *managing relations* domain, which focuses on the interaction between various stakeholders, organizations, and levels of government involved in policy implementation, is reflected in the need for coordinated, cross-departmental efforts and clear departmental responsibilities to effectively manage these risks.

Government departments   Managing departmental roles and expectations in response to extreme heat emerged as a key challenge. Participants emphasized that, in addition to local action plans and preventative measures during high temperatures, clear roles and responsibilities would help effective management of heat-related health risks. A municipal participant expressed that guidance could be helpful to address heat, as the responsibility may sit outside of one single department.

"*Heat is something else completely, because there's no single department in the city that is responsible for heat planning, because it's not a traditional function of local government*" P012, Municipal, Non-Health Sector.

In the Metropolitan municipality City of Cape Town, climate change action and planning is led by the City's resilience department. A municipal participant highlighted that a cross-cutting department like this would be best positioned to lead heat planning, which also ensures coordination between climate action and heat response efforts (P012).

At the provincial level, the Disaster Risk Management Center (DRMC) of the Western Cape is seen as a key department in addressing such cross-cutting risks. Participants noted that the DRMC identifies high-risk areas for heat and provides guidance on risk reduction (P002; P008; P010; P029). This includes issuing information to municipalities in the Western Cape about high temperature days and the potential risk of fires, helping to raise community awareness of health impacts and mitigate risks, such as by restricting open burning.

"*They [disaster management] normally indicate to us when it's very hot days. And uh where there's possibilities of, say heat stroke and things like that for the public and the communities in summer months, they do send out notifications SMS and emails and things like that.*" P029, Municipal, Non-Health Sector

Participants, particularly at the provincial level, emphasized the importance of building partnerships with cross-departmental stakeholders working on heat-related issues (P002; P008; P012; P015). They highlighted the need for information sharing and coordinated responses, despite the absence of a provincial heat action plan. The provincial Department of Environment initially intended to collaborate with DRMC to develop such a plan. However, as expressed by a provincial environment sector participant, concerns arose that their role might mirror their experience with the Western

Cape Climate Change Strategy, where they primarily developed the plan that required constant advocacy for engagement among other government departments, such as the provincial Department of Health (P002). As a result, the Department of Environmental Affairs and Development opted for a different approach, choosing to focus on communication and awareness as key components of heat action planning.

During interviews, it was discovered that national government supports municipalities directly with managing heat risks through resources such as a Heat-Health Risk Vulnerability Assessment tools and the *Let's Respond* toolkits, a resource to support municipal climate change planning integration (P007; P014; P028; P029). The experience from municipal participants was that these tools and planning strategies, especially related to heat, needed to be adapted to ensure their practical application at the local level.

*"National Department assisted us, they assisted all district municipalities in in the country with, Let's respond to Heat…the National Department assisted us with getting a draft like a draft plan for the West Coast District and we made it part of our own and we adapted it a little bit to make it part of the West Coast District."* P029, Municipal, Non-Health Sector

Finally, challenges related to limited human and financial resources, along with an unstable political environment, significantly impacted heat mitigation activities at the municipal and district levels. Interview participants noted that municipalities often lack the human and technical capacity to effectively respond to heat events, specifically difficulties in recruiting skilled workers for facility retrofitting (P004; P025; Po26; P029). Furthermore, there were times when insufficient funds impeded efforts to support retrofitting and provide consistent public messaging during heatwaves.

Health facilities When it came to heat-health activities within health facilities and clinics, several provincial health participants highlighted ongoing efforts to improve health facility infrastructure to mitigate heat-related health risks (P004; P008; P025; P026). These efforts include upgrading older facilities to better withstand extreme temperatures, both heat and cold, and incorporating temperature-regulating structures into newly constructed health clinics and buildings. One provincial health sector participant noted that newer facilities are integrating design features that optimize airflow and heat management (P026). However, implementing similar innovations in older facilities is often impractical due to the high costs of retrofitting them effectively.

*"So, I think we gradually getting there, but we are sitting with a lot of, of a very old buildings where the only way to, you know, heat them up in in in winter or cool them down in summer is to is to run air conditioners, which of course, use a lot of power and so on… I guess there's a much more thought tied into the style than they used to be. But unfortunately, we've got the legacy a little bit like asbestos, we put the legacy of 100 years of bad design. And, and it's extremely expensive to replace hospitals and clinics."* P026, Provincial, Health Sector

Beyond the cost challenges of adapting existing health facilities, municipal and provincial participants shared that heat adaptation strategies for health facilities and clinics, proposed by national and provincial level, were often not adapted to the local context, leading to significant implementation issues. A provincial health participant noted that retrofitting waiting rooms frequently faced obstacles due to what was termed a "one-size-fits-all" approach, often imposed by the provincial level. Participants who work closely with municipalities reported that building infrastructure plans for addressing heat were applied uniformly across the province rather than being customized to specific local needs and conditions (P025; P026). As a result, this standardized approach often led to ineffective strategies or even created additional challenges, as observed in health facilities in the Central Karoo.

*"If you look at the designs of our clinics and hospitals, there's a generic model that is applied and this model is applied province wide. This model doesn't necessarily consider extreme outliers in terms of hot and colder, it looks at a more*

*generic temperature average across the province….and one of the issues that we do have is that our model doesn't necessarily accommodate these [extremes], which we do find a bit of a challenge,* P025, Provincial, Health Sector

Some municipal participants further noted that a major challenge in adapting heat strategies is the disconnect between local experiences and policy expectations. They highlighted frequent conflicts with national and provincial policymakers over how facilities should be designed versus what works best in the local environment (P025; P029). In this instance, the provincial level could step in to mediate and as one participant suggested, provincial health department could play a key role in facilitating cross-learning between hospitals that have already implemented heat adaptations (P005).

At the directional level, the *context maintenance* domain highlights how provincial and municipal departments are best positioned to adapt policy strategies and planning to local realities, ensuring proper implementation. At the operational level, the *managing relations* domain emphasizes the importance of clearly defined departmental roles and inter-departmental collaboration, both of which are crucial for addressing heat-related health risks.

**Individual level and perceptions.** At the individual level, how heat-risks are perceived, as well as participant's past experiences have an influence on how heat-related health risks are managed. The governance domains *internalization of values and norms* refers to the ways in which stakeholders absorb societal expectations and organizational norms that shape their decision-making, while *situation-bound rule application* emphasizes how individuals apply policies to specific circumstances based on the situation at hand. Most provincial and municipal participants were aware of the health impacts of increased heat and discussed actions that were taken.

A critical influence on the governance of heat-related health risks is the extent to which stakeholders perceive heat as a risk, which varied across provincial and municipal participants. Many expressed being accustomed to high temperatures that could lead to inaction.

"*I think heat waves we probably don't take as seriously, because we experienced them so frequently. You know, being in Africa.*" P023. Non-Health Sector, Municipal

Although there was an awareness and understanding of health impacts associated with increasing temperatures, especially among health participants, when it came to other immediate or competing risks, heat was not seen as much of a concern. This was expressed by a provincial health sector participant:

"*I think there's been less emphasis on heat… in terms of responding or manage, I mean, I don't think we've experienced that. I mean, I think during our summer months, we always have days that are quite hot. And there are people who are affected directly, as you say, with heat stroke. But it hasn't been a major issue that we've had to deal with.*" – P027 Provincial, Health Sector

Several provincial participants shared that they did not see heat as an issue and that communities were also unaware of heat as an issue that needs to be addressed.

"*I don't know what you're referring to in terms of heat wave because I mean, I, you know, I don't think it was enough to register in people's consciousness. I live in Cape Town and I don't see a distinctive Heatwave, I'm sure we have problems with some heat somewhere. But there's nothing compared with like what happened to them in Vancouver last year as an example. So, the question is, how do we take heat?*" - P30 Provincial, Non-Health Sector

In fact, some health sector participants and several non-health sector participants across all governance levels saw the mitigation of heat risks fully as the responsibility of individuals, where the most that departments and health workers could be responsible for is risk communication (P001; P002; P0016; P027; P030). Participants noted that citizens and civil

society share a responsibility, in this case with the health workforce, for reducing exposure to heat. This was especially highlighted by a national, non-health sector participant who emphasized that the sharing of information, such as high heat days, was sufficient for an individual to understand they are at high risk and able to take the appropriate measures to protect themselves.

"*People getting to know that when you are elderly or you are younger than five and all of that you need to be staying away from the sun, making sure that you drink enough water and all of those things and have your standby or emergency services up and ready for such responses and many other things*" P016, National, Non-Health Sector

Provincial and municipal participants shared that health promotion efforts do occur, especially by environmental health practitioners and health promotion officers. For example, heat warnings and information are shared with citizens and communities, especially the most vulnerable, to mitigate individual heat risks. Within health facilities and clinics individual efforts vary. One provincial participant discussed that in the region they oversee, there is clinic-based monitoring of heat stroke and related issues, and if there are several cases in one area, they are reported to municipality (P031). There was also discussion of increased vigilance by health care staff during periods of high temperatures, typically from November to May, due to an observed increased incidence of diarrheal disease. Additionally, on high heat days health facility staff are advised to come in earlier and leave later to avoid the heat (P031), which is an example of health workforce protection and adaptation to heat.

While healthcare and public health staff are aware of health risks during periods of extreme heat, participants discussed the challenges they face due to a perceived lack of control in managing heat. A provincial health participant expressed that despite existing efforts to inform of high heat days, people still engage in high-risk behaviors, like continuing an outside activity or scheduled sport event.

"*There was not a lot of readiness because people like continued working their normal hours. There was even I think there were competitions that took place so people, older learners, going outside, doing athletics physical work, while it was quite warm. So clearly, people had not actually taken that precaution*" P002, Provincial, Non-Health Sector

At the individual and perception level, *internalization of values and norms* emphasizes how the perception and understanding of heat risks can impact the decisions to manage heat-health strategies, while *situation-bound rule application* emphasizes how individuals may prioritize heat-health actions depending on their observations and the circumstances at hand, such as changing behaviors due to a known high heat day.

Table 3 summarizes participants' perspectives on managing heat-related health risks, organized by governance domains and action scales identified in the findings.

## Discussion

To our knowledge, this is the first study to assess the implementation challenges and enablers of the Heat Health Action Guidelines in South Africa, especially at a subnational and local level. Overall, the findings highlight that despite the existence of a heat-health plan and recognition from Western Cape decision-makers of the importance of managing heat-related health risks, implementation remains fragmented. While the provincial and municipal stakeholders, both within and beyond the health sector, are making concerted efforts to mitigate the impacts of extreme heat on public health, these efforts are not fully aligned with heat-health policy and uncoordinated.

This study has several limitations. First, it formed part of a larger research project that also examined drought as a climate risk. Therefore, heat was not the sole focus of inquiry, which may have constrained the type and depth of information gathered on heat risks. Second, the findings rely on in-depth interviews that reflect participants' subjective experiences and perspectives. This introduces the possibility of recall bias bias, and the insights may not be representative of the wider

**Table 3. Summary of Findings for Managing Heat-Related Health Risks by Domains of Governance.**

| Acts of Governance | | Experiences Managing Heat-related Health Risks |
|---|---|---|
| SYSTEM | General rule setting | • Viewing heat as a common threat supports coordination across sectors to address heat-related health risks<br>• Limited awareness of 2020 National Heat Health Action Guidelines |
| | Managing trajectories | • Innovative approaches required to address heat can arise when engaging with diverse set of stakeholders and the community<br>• Plans for a multisector heatwave index can support local heat action and planning |
| ORGANIZATIONAL | Context Maintenance | • Adapting strategies and tools, often provided as a "one-size fits all approach" from national and provincial level, at a local level is critical for effective implementation<br>• Costs have hindered needed renovations to health facility infrastructure to mitigate heat risks |
| | Managing Relations | • A lack of clear roles and responsibilities for departments and staff impacts effective management of heat risks<br>• A cross-cutting department could take on heat planning and coordinate cross-departmental partnerships essential to manage heat-health activities |
| INDIVIDUAL ⇕ PERCEPTION | Internalization of values & norms | • Perceived lack of control in managing heat due to community and individuals engaging in high-risk behaviors<br>• Variation in perceived risk of heat by decision makers and the public can impact action taken to address heat-related health risks |
| | Situation-bound rule application | • Health promotion and some monitoring of heat risks by health workers in clinics occurring, especially at observed times of increased risk (November to May)<br>• Perceived competing priorities minimize need to address heat risks |

population. Although the sample size and selection methods were appropriate for exploratory analysis, selection bias remains a potential limitation, as those who agreed to participate may differ from those who did not. Finally, to preserve participant anonymity, distinctions between departments could not be explored, limiting the ability to examine how organizational contexts may shape the challenges faced and the actions taken to manage heat-related health risks. Collectively, these factors restrict the generalizability of the findings, though they provide valuable insights into emerging themes and areas for further research.

Our findings suggest that implementation opportunities and challenges occur across multiple action scales. At the systems level, coordination across sectors to respond to heat risks and develop innovative approaches emerged as critical enablers. At the organizational level, adapting existing strategies to the local context and clear departmental roles and responsibilities is essential for managing heat risks effectively. However, challenges such as adapting general strategies to local settings and high costs of retrofitting facilities were also highlighted. At the individual level, perceptions of heat risks influence the urgency of action, with competing priorities and a perceived lack of control contribute to delays in addressing heat-related health threats. While acknowledging the interconnectedness of all action scales and levels, our findings indicate that the organizational and individual levels, more than others, surfaced the greatest number of issues with significant implications for successful implementation. Managing heat-related health risks relies on coordination across sectors, clarity of departmental roles, local adaptation of policy strategies, and perception of heat risks.

## Coordinating across sectors (system level)

Our findings suggest that participants look to system level guidance to support coordination efforts. Such guidance can take the form of clarifying roles and responsibilities or proposing flexible strategies that can be practically operationalized at the local level. As theoretical work on multilevel governance and policy processes emphasizes, nested rules affect outcomes in any setting, as all levels are always implicitly interconnected [18,30,31]. The MGF framework reflects this, illustrating how the constitutive governance level allows directional and operational level actions within the system level action scale. In the context of heat risk management, action follows a process that relies on direction and support from macro-level, in this case, the national level and their development of the Heat Health Action Guidelines.

Although the Action Guidelines were developed by the National Department of Health, the cross-cutting nature of heat risks means multiple sectors are affected. Heat is often framed primarily as a health issue, which can lead other sectors to defer responsibility to the health sector for addressing heat-related risks. A recent study highlights that such framing may limit cross-sectoral awareness and collaboration in addressing the broader impacts of heat [32]. Here the concepts raised by Shiffman and Smith [29] regarding the framing, understanding, and portrayal of an issue can also influence the extent to which actors find solutions to the problem. This can also suggest that when heat is framed solely as a health risk, it may affect stakeholder buy-in and ultimately hinder cross-sectoral coordination and implementation. Despite these challenges, participants noted that during heatwaves, multiple sectors came together to respond, recognizing heat as a singular and urgent threat. This highlights the potential for cross-sectoral collaboration during crises but sustained and coordinated action beyond emergencies is necessary to manage heat-related health risks effectively.

### Clear departmental roles and responsibilities (organizational level)

Clarifying departmental roles and responsibilities is essential for ensuring accountability in heat risk management. A key barrier to local climate adaptation in South Africa is the perception that environmental and climate-related issues fall solely under the responsibility of the provincial environment department and environmental health practitioners (EHPs) who work for municipalities [33,34]. Although participants in this study did not identify heat mitigation and climate adaptation as the responsibility of EHPs, there was acknowledgement that EHPs play an important role in behavior change and health promotion at the municipal level. Clarifying roles amongst departments and institutions is critical to action. In a study in Germany by Geffert et al. [11] the implementation of heat health action plan at the subnational and local level by the public health service, and no other sectors, was examined. Their analysis of policy documents and interviews revealed that heat was not adequately mentioned in subnational frameworks and roles were not clearly defined to support implementation of the German Heat-Health Action Plans [11].

Similarly in a policy document analysis looking at climate and health policy in the Western Cape, documents not do not clarify specific departmental roles and responsibilities to address heat [25]. Unlike the City of Cape Town, where the resilience department has since taken on heat planning and laying out roles and responsibilities among municipal departments, the Western Cape Department of Environment has not assumed a similar role despite coordinating the Provincial climate change strategy development, leaving the province without a dedicated heat action plan. While Western Cape participants in this study suggested that the Provincial Disaster Risk Management Center would be the most suitable department to coordinate heat action planning, this has not yet materialized to date. Even if another sector led coordination of heat action, elsewhere in South Africa Nana *et al.,* 2019 argue that unclear institutional mandates and a lack of transparency around heat and adaptation planning leads to limited action on heat-related health risks in the City of Johannesburg [35]. An assessment across multiple cities in India similarly found significant issues with preparedness for longer term heat risks related to institutional challenges, including how heat plans are institutionalized [36]. Yet, in efforts reported at the subnational level in India, States like Uttar Pradesh have a Heat Action Plan with a framework for implementation developed by the Uttar Pradesh State Disaster Management Authority to mitigate health risks from rising temperatures, and this is seen to work well [37].

### Adapting to the local context (organizational level)

The limited involvement of local stakeholders in policy and strategy development can further hinder effective implementation. Many participants in this study expressed a lack of awareness about the Action Guidelines, suggesting minimal involvement of local stakeholders in the development of the policy. This is not uncommon in heat-related health planning. As Vanderplanken et al. [12] point out, heat-health action planning face significant challenges due to limited engagement from local authorities. This challenge is further compounded by the fact that national heat-health plans are often developed as "non-committal guidelines," that place a heavy reliance on the prioritization and motivation of local authorities to drive implementation ([12], p. 10).

Our findings emphasize the importance of avoiding overly prescriptive national policies and emphasize that involving subnational and local decision-makers in policy development helps ensure that strategies are flexible and adaptable to local contexts. For heat-health strategies to be successfully implemented at the local level, it is essential to consider local contexts and associated costs, such as those necessary for upgrading clinic and hospital infrastructure. Since the interviews were conducted in early 2022, the national department has made progress in bringing the Action Guidelines to the local level and improving strategies. This is particularly important as several provincial and municipal participants reflected on the challenges of applying national strategies to local contexts. Ravishankar et al. [32] underscore that tailoring strategies to local realities and establishing well-functioning institutional arrangements are foundational for managing heat risks, particularly immediate ones. Regular inclusion of local stakeholders in institutional processes could facilitate the flow of local data to inform and refine future heat-risk strategies, ultimately strengthening local adaptation efforts.

**Perception of heat risks (individual level)**

At the individual level, stakeholder perceptions and experiences play a critical role in heat-health governance. Heat impacts on health often go unnoticed due to the difficulty in attributing symptoms or causes of death to heat stress, and this challenge is compounded by the perception that heat is more tolerable in hot regions, leading to a dismissal of risks and a diminished sense of urgency to address them [38]. This perception was evident in a South African sub-district community where although study participants acknowledged vulnerabilities associated with extreme heat, they expressed acclimating to increasing temperatures [39]. Similar findings emerged in studies conducted in high-income countries, including Scotland and England, where a perceived low risk delayed the prioritization of crucial actions and preventative measures to address heat-related health threats [40,41]. In Wan et al.'s [40] study of heat-health governance in Scotland, interview participants reported a lack of perceived heat risks due to historically cooler weather, a preference for warm days, and the belief that they were not vulnerable to heat effects. Lo [42] further highlights that differential perceptions of risk can be a key barrier to adaptive behavioral adjustments, such as purchasing flood insurance. The study concluded that adaptive actions are not only influenced by risk perception but also by their impact on the functioning of social networks and institutions [42]. Consequently, individual norms and behaviors among stakeholders can significantly influence community adaptation to climate change and, in this case, to heat risks.

As this study and previous literature highlights, the absence of clearly defined roles and responsibilities, essential for effective implementation of heat-health strategies, makes decision-makers' perceptions of heat risks more critical in shaping responses and driving action to manage heat-related health risks. A shift in perception, coupled with clearer institutional arrangements and cross-sectoral collaboration, could significantly enhance the implementation of heat-health action plans and improve resilience to rising temperatures.

## Conclusion

This study offers novel insights into the implementation challenges and enablers of the Heat Health Action Guidelines in South Africa, with a particular focus on subnational and local levels. Our findings suggest that effective implementation requires coordinated cross-sectoral action, clear departmental roles, and local adaptation of policy strategies. However, challenges such as limited local stakeholder engagement, high costs of retrofitting infrastructure, and varying perceptions of heat risks continue to hinder progress. Addressing these challenges calls for a shift toward inclusive policy development that involves local decision-makers, establishes clear institutional mandates, and fosters sustained cross-sectoral collaboration beyond crisis periods. Strengthening these dimensions will lead to more effective heat-health action, reducing the impacts of extreme heat events on public health and improving the capacity of subnational and local stakeholders to manage future heat-related challenges.

## Supporting information

**S1 File. Full Master Interview Guide.**
(DOCX)

## Acknowledgments

A very special thank you to the research participants who contributed to the findings of this study.

## Author contributions

**Conceptualization:** Amanda Victoria Quintana, Susannah H. Mayhew.

**Data curation:** Amanda Victoria Quintana.

**Formal analysis:** Amanda Victoria Quintana.

**Funding acquisition:** Amanda Victoria Quintana.

**Investigation:** Amanda Victoria Quintana.

**Methodology:** Amanda Victoria Quintana.

**Project administration:** Amanda Victoria Quintana.

**Resources:** Amanda Victoria Quintana.

**Supervision:** Lucy Gilson, Sari Kovats, Susannah H. Mayhew.

**Validation:** Lucy Gilson, Caradee Y. Wright, Susannah H. Mayhew.

**Visualization:** Amanda Victoria Quintana.

**Writing – original draft:** Amanda Victoria Quintana.

**Writing – review & editing:** Amanda Victoria Quintana, Lucy Gilson, Sari Kovats, Caradee Y. Wright, Susannah H. Mayhew.

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
