## [Decision Letter · Decision Letter 0]

17 Jun 2025

PGPH-D-25-01143

The Implementation Gap: Cross-Sector Management of Heat-related Health Risks in Western Cape, South Africa

Dear Dr. Quintana,

Thank you for submitting your manuscript to PLOS Global Public Health. After careful consideration, we feel that it has merit but does not fully meet PLOS Global Public Health’s publication criteria as it currently stands. Therefore, we invite you to submit a revised version of the manuscript that addresses the points raised during the review process.

We look forward to receiving your revised manuscript.

Kind regards,

Bhargav Krishna

Academic Editor

Journal Requirements:

Additional Editor Comments (if provided):

Thank you to the authors for the opportunity to review this important manuscript. The results are fascinating and could be an important contribution to the heat-health implementation literature in a re-worked form. The reviewer comments focused on a few areas where the manuscript requires significant strengthening and I would urge the authors to revise and resubmit their manuscript with these concerns addressed.

1. Engagement with the broader literature on heat response globally and implementation of heat action plans

2. The use of the framework and whether it serves its role in eliciting the broad range of challenges in implementation

3. The use of COREQ criteria to strengthen the structural aspects of the manuscript

4. Clarifying the governance structures involved for a broader readership

5. Highlighting the limitations of the work more clearly

Reviewers' comments:

Reviewer's Responses to Questions

**Comments to the Author**

1. Does this manuscript meet PLOS Global Public Health’s publication criteria?

Reviewer #1: Yes

Reviewer #2: Yes

2. Has the statistical analysis been performed appropriately and rigorously?

Reviewer #1: N/A

Reviewer #2: Yes

3. Have the authors made all data underlying the findings in their manuscript fully available (please refer to the Data Availability Statement at the start of the manuscript PDF file)?

Reviewer #1: Yes

Reviewer #2: Yes

4. Is the manuscript presented in an intelligible fashion and written in standard English?

Reviewer #1: Yes

Reviewer #2: Yes

Reviewer #1: Topic:

The Implementation Gap: Cross-Sector Management of Heat-related Health Risks in Western Cape, South Africa

Journal: PLOS Global Public Health

Date: 20th March 2025

Abstract:

The abstract matches the title well in terms of focus and scope.

I looked out for procedural clarity, and methods and process are clearly laid out. E.g.

• Data Source: 31 in-depth interviews.

• Triangulation: Cross-referenced with prior policy document analysis.

• Analytical Framework: Adapted Multiple Governance Framework.

• Levels of Analysis: Provincial and municipal levels, cross-sector stakeholders.

• Outcome: Identification of governance gaps, implementation barriers, and recommendations.

Thus, the procedural description is clear, credible, and appropriate for a governance-focused study.

I would suggest a sentence structure tightening: For example, the last sentence is long and slightly confusing "decisionmakers in the Western Cape can manage current and future heat-related health risks that can communities are better equipped..." This is likely a typo or syntax error ("that can" should be removed or rephrased).

Introduction:

For this, I looked out for existing knowledge (which was clearly established). What is unclear is a clear, explicit statement of the research gap. I rather saw an inferred gap (what it seems to be):

Although South Africa has formally recognized heat as a health issue and developed action guidelines, it is unclear how these plans are implemented at the subnational/local level, particularly in terms of cross-sectoral coordination, and whether the top-down approach limits effective governance at those levels.

Clearly state what is missing in existing literature and how the current study addresses that to improve readability

Methods:

I assessed this part using the COREQ (Consolidated Criteria for Reporting Qualitative Research) checklist, and that leaves a lot of gaps which could improve. E.g.

• Almost all Domain 1 (Research Team and Reflexivity) items are missing.

• Little detail on researcher-participant relationship, recording, field notes, or participant validation.

• Data analysis team details (number of coders, how consensus was reached) are absent.

• No mention of data saturation or whether interview guides were piloted or reviewed.

Therefore, while methodologically solid in its sampling and analysis descriptions, the methods section lacks crucial reflexivity and transparency elements common in high-quality qualitative research reporting. Strengthening these areas would improve rigor and replicability.

Results:

The study aimed to understand how heat-related health risks are managed in the Western Cape, especially at subnational and local levels, and why an implementation gap exists despite national guidelines. The themes are well aligned with this aim and reflect the core focus. The conceptual framing is also appropriate and methodologically sound.

The discussion reiterates the research gap: lack of implementation analysis at subnational levels in South Africa. It reinforces the study’s unique contribution while building on prior studies that framed implementation as overly top-down. This successfully addresses the gap hinted at in the introduction.

Discussion:

The discussion does not explicitly mention limitations of the study (e.g., generalizability, possible biases from sampling, reliance on self-reported action).

There’s no mention of future research directions or how the findings might be applicable beyond Western Cape.

Q1. Is the manuscript technically sound, and do the data support the conclusions?

Yes, the manuscript is technically sound, and the data support the conclusions within the context of a qualitative study. Also, the conclusions are clearly supported by the thematic results. The findings around awareness, system-level coordination, organizational bottlenecks, and individual perceptions all feed logically into the conclusion. The authors are careful not to overgeneralize. Their conclusions focus on the Western Cape context and acknowledge variation across governance levels and sectors. Minor improvements could further strengthen transparency and rigor (COREQ Checklist).

Q2. Has the statistical analysis been performed appropriately and rigorously?

Since this is a qualitative study, statistical analysis is not applicable. However, I assessed the rigor of the qualitative analysis, which functions as the analytic equivalent in such research. Therefore, I looked at the appropriateness of method used, the framework, coding process, software, transparency and procedural clarity. If COREQ is applied, it will enhance the rigor.

Q3. Have the authors made all data underlying the findings in their manuscript fully available?

Only the interview guide is provided. They did state that their data is also provided, although it is not seen here

Q4. Is the manuscript presented in an intelligible fashion and written in standard English?

The manuscript is presented in an intelligible fashion and written in standard English. However, there are some minor errors. E.g. the last sentence of the abstract earlier mentioned, Lass et al., 1375" seems to be a typographical error so verify and correct the year (it’s likely not 1375), and also minor sentence structuring issues.

Reviewer #2: Comments to the Authors

This paper offers important insights into the fragmented governance of heat-related health risks in South Africa’s Western Cape province. It shows how limited local engagement, unclear institutional mandates, and weak coordination across sectors have contributed to the poor uptake of the national Heat-Health Action Guidelines. The topic is timely and highly relevant, especially as climate change intensifies heat-related health risks in many low- and middle-income countries. The empirical work is rich and well-grounded in the South African policy context. However, the paper’s contribution could be strengthened by clarifying the research framing, improving the transparency of methods, and making the analysis more accessible to international readers. I offer the following suggestions to improve the paper’s clarity, rigour, and broader relevance.

Major Suggestions

1. The research question should be stated more clearly. While the paper outlines its focus, it does not explicitly frame a research question. A short and direct research question at the end of the introduction, ideally highlighting the aim to identify barriers and drivers of action, would help orient the reader and clarify the analytical direction.

2. The paper explains the sampling and data collection approach, but it does not describe how thematic saturation was assessed. Explaining when and how saturation was reached would improve the credibility of the findings. If saturation was not a central concern, the authors could acknowledge this explicitly or note it as a limitation.

3. The paper could reflect more critically on the use of the Multiple Governance Framework. The framework is useful for organising the findings, but applying it across all levels of analysis seems to limit the exploration of more complex or cross-cutting themes. For example, the analysis at the organisational level does not fully explore interdepartmental tensions or differences in institutional capacity. The authors could consider whether certain insights might be better presented outside the framework to allow for a fuller discussion.

4. The use of quotes could be clarified. In some places, it is difficult to tell whether a quote reflects a commonly held view or a singular perspective. Where possible, the authors should indicate how many respondents expressed similar views. This would help strengthen the link between the qualitative data and the broader findings.

5. The paper would benefit from a visual summary of sectoral roles. Since one of the main findings is that implementation is fragmented across departments, a table or diagram showing which departments are involved, what actions they are taking (or not taking), and how these align with the national guidelines would help make the argument more compelling. It could also help establish the need for cross-sectoral action earlier in the paper. A visual summary of heat-related actions mapped to various departments (even from comparable cities if Western Cape data is limited) would further support this case.

6. The paper could engage more deeply with comparative literature from other heat governance work. India’s experience with national and subnational heat action planning offers valuable points of comparison. For instance, Pillai’s recent work on heat action in India (Pillai, A. V., T. Dalal, I. Kukreti, A. Kassinis, L. V. Zeppetello, E. Tewari & N. K. Dubash. (2025). Is India ready for a warming world? How heat resilience measures are being implemented for 11% of India’s urban population in some of its most at-risk cities. Report. New Delhi.) presents findings that closely align with those in this paper and could be used to draw parallels. The paper would also benefit from referencing literature on heat governance that goes beyond the health sector. Relevant works include:

Meerow, S., & Keith, L. (2021). Planning for Extreme Heat: A National Survey of U.S. Planners. Journal of the American Planning Association, 88(3), 319–334.

Bolitho, A., & Miller, F. (2017). Heat as emergency, heat as chronic stress: Policy and institutional responses to vulnerability to extreme heat. Local Environment, 22(6), 682–698.

Howarth, C., McLoughlin, N., Armstrong, A. et al. (2024). Turning up the heat: Learning from the summer 2022 heatwaves in England to inform UK policy on extreme heat. Grantham Research Institute on Climate Change and the Environment, LSE.

7. The paper assumes a level of familiarity with South Africa’s institutional context that may limit its accessibility to international readers. A short explanation of the 2020 Heat-Health Action Guidelines, a brief overview of South Africa’s three-tier governance structure, and a sentence or two explaining what Heat Action Plans typically include would help orient readers who are not familiar with the country or the field.

Minor Suggestions

1. Some quotes appear cut off or lack context, particularly the one on Line 282. Please review all quotations for clarity and completeness.

2. Line 35 contains a grammatical error (“that can communities are...”) and should be revised. Line 610 should read “system-level.”

3. A table comparing barriers faced by health and non-health departments could help highlight institutional differences in engagement and capacity. This is completely optional, but could help make this more policy oriented.

I am also attaching a PDF of the manuscript with a few additional minor comments and clarification suggestions.

**Do you want your identity to be public for this peer review?** For information about this choice, including consent withdrawal, please see our Privacy Policy

Reviewer #1: No

Reviewer #2: **Yes: ** Tamanna Dalal

---

## [Decision Letter · Decision Letter 1]

3 Oct 2025

The Implementation Gap: Cross-Sector Management of Heat-related Health Risks in Western Cape, South Africa

PGPH-D-25-01143R1

Dear Ms,. Quintana,

We are pleased to inform you that your manuscript 'The Implementation Gap: Cross-Sector Management of Heat-related Health Risks in Western Cape, South Africa' has been provisionally accepted for publication in PLOS Global Public Health.

Best regards,

Lina Taing

Academic Editor

Reviewer Comments (if any, and for reference):

Reviewer's Responses to Questions

**Comments to the Author**

Reviewer #1: All comments have been addressed

Reviewer #2: All comments have been addressed

publication criteria?

Reviewer #1: Yes

Reviewer #2: Yes

3. Has the statistical analysis been performed appropriately and rigorously?

Reviewer #1: N/A

Reviewer #2: Yes

4. Have the authors made all data underlying the findings in their manuscript fully available (please refer to the Data Availability Statement at the start of the manuscript PDF file)?

Reviewer #1: Yes

Reviewer #2: Yes

5. Is the manuscript presented in an intelligible fashion and written in standard English?

Reviewer #1: Yes

Reviewer #2: Yes

Reviewer #1: (No Response)

Reviewer #2: Thank you for addressing the previous comments. The new addition to the limitations section covers a lot of questions I had as a reader.

**Do you want your identity to be public for this peer review?** For information about this choice, including consent withdrawal, please see our Privacy Policy

Reviewer #1: No

Reviewer #2: No
